# Long-Term Nightshift Work and Breast Cancer Risk: An Updated Systematic Review and Meta-Analysis with Special Attention to Menopausal Status and to Recent Nightshift Work

**DOI:** 10.3390/cancers13235952

**Published:** 2021-11-26

**Authors:** Christine Schwarz, Ana María Pedraza-Flechas, Roberto Pastor-Barriuso, Virginia Lope, Nerea Fernández de Larrea, José Juan Jiménez-Moleón, Marina Pollán, Beatriz Pérez-Gómez

**Affiliations:** 1Public Health and Preventive Medicine Teaching Unit, National School of Public Health, Instituto de Salud Carlos III, Avda. Monforte de Lemos 5, 28029 Madrid, Spain; cschwarz@isciii.es; 2Medicine and Health Sciences, Universidad del Rosario, Calle 12C No 6-25, Bogotá D.C. 111221, Colombia; anama.pedraza@urosario.edu.co; 3Department of Epidemiology of Chronic Diseases, National Centre for Epidemiology, Instituto de Salud Carlos III, Avda. Monforte de Lemos 5, 28029 Madrid, Spain; rpastor@isciii.es (R.P.-B.); vicarvajal@isciii.es (V.L.); nfernandez@isciii.es (N.F.d.L.); mpollan@isciii.es (M.P.); 4Consortium for Biomedical Research in Epidemiology and Public Health (CIBER en Epidemiología y Salud Pública—CIBERESP), 28029 Madrid, Spain; 5Department of Preventive Medicine and Public Health, School of Medicine, University Granada, Avda. de la Investigación No 11, 18016 Granada, Spain; jjmoleon@ugr.es

**Keywords:** nightshift work, breast cancer, recent exposure, meta-analysis, menopausal status, occupational exposure, retirement age

## Abstract

**Simple Summary:**

Night shift work (NSW) may disturb circadian rhythms. This could influence the risk of breast cancer (BC), but research papers have reported conflicting results. We reviewed, summarized, and combined the results of those studies that measured the effect of long-term nightshift work (≥15 years of NSW) in BC with special attention to menopausal status and time since retirement age. Women with long-term NSW had 13% more risk of BC than women without NSW. Postmenopausal women showed no increased risk, while premenopausal women had a 27% higher risk. Women with a higher probability of recent long-term NSW (women under retirement age) had a 23% higher risk than women without NSW. We concluded that long-term NSW may increase BC risk, especially in women before menopause or shortly after NSW discontinuation.

**Abstract:**

This systematic review discusses long-term NSW and female BC risk, with special attention to differences between pre- and postmenopausal BC, to test the association with recent NSW. The review follows PRISMA guidelines (Prospero registry: CRD42018102515). We searched PubMed, Embase, and WOS for case–control, nested case–control, and cohort studies addressing long-term NSW (≥15 years) as risk exposure and female BC as outcome until 31 December 2020. Risk of bias was evaluated with the Newcastle–Ottawa scale. Eighteen studies were finally included (eight cohorts; five nested case–control; five case–control). We performed meta-analyses on long-term NSW and BC risk; overall and by menopausal status; a subanalysis on recent long-term NSW, based on studies involving predominantly women below retirement age; and a dose–response meta-analysis on NSW duration. The pooled estimate for long-term NSW and BC was 1.13 (95%CI = 1.01–1.27; 18 studies, I^2^ = 56.8%, *p* = 0.002). BC risk increased 4.7% per 10 years of NSW (95%CI = 0.94–1.09; 16 studies, I^2^ = 33.4%, *p* = 0.008). The pooled estimate for premenopausal BC was 1.27 (95%CI = 0.96–1.68; six studies, I^2^ = 32.0%, *p* = 0.196) and for postmenopausal BC 1.05 (95%CI = 0.90–1.24,I^2^ = 52.4%; seven studies, *p* = 0.050). For recent long-term exposure, the pooled estimate was 1.23 (95%CI = 1.06–1.42; 15 studies; I^2^ = 48.4%, *p* = 0.018). Our results indicate that long-term NSW increases the risk for BC and that menopausal status and time since exposure might be relevant.

## 1. Introduction

Many women’s working lives are characterized by some degree of night work, either as permanent shifts or as shifts within rotating work schedules. The European Union defines “night time” as any period of seven or more hours- as defined by national law, which includes the period between midnight and 5.00 a.m. [1]. Two large surveys provide information about the prevalence of night work in western societies [2,3]. According to the “Sixth European Working Conditions Survey” (2015), 14% of working women carried out night work [3], while in the “American Time Use Survey” of the US Bureau of Labor Statistics (2017/2018), 3.9% of working women reported night work, 2.1% rotating shifts, and 2.5% irregular shifts [2].

Humans share with other organisms the fundamental properties of a biological circadian system [4]. Night work has been studied as a possible risk factor for several diseases because physiological functions such as temperature and glucose homeostasis, hormone secretion, and immunological functions follow endogenous circadian rhythms [5]. These rhythms are controlled by the central circadian pacemaker, located in the suprachiasmatic nucleus of the hypothalamus [4]. Cellular circadian clocks generate roughly 24 h oscillations in the function of thousands of genes, allowing the organism to anticipate regular changes, to optimize energy, and to temporally separate incompatible processes [6]. As light is the most important stimulus that may reset circadian rhythms daily, by direct neuronal connections between the retina and the pacemaker, mistimed exposure to artificial light, usual in nightshift work (NSW), can cause important changes in the phase and amplitude of circadian rhythms, with deleterious effects in health [7].

In 2019, the International Agency for Research on Cancer, confirming a previous evaluation, classified NSW as probably carcinogenic to humans (Group 2A on a scale with a range from 1 “carcinogenic to humans” to 3 “not classifiable as to its carcinogenicity to humans”). Its decision was based on limited evidence from studies in humans and strong evidence from experimental findings, which showed its effects in immunosuppression, chronic inflammation, and cell proliferation [8]. The tumors with positive associations with NSW were mainly breast, prostate, and colorectal cancer [8].

This issue has awakened enormous interest, given the global burden of breast cancer (BC) [9] and the considerable prevalence of NSW. Unfortunately, the results of individual epidemiological studies have so far been inconsistent. Additionally, a number of meta-analyses have come to different conclusions [10,11,12]. One factor that could be related to the inconsistency of results in observational studies is the duration of NSW. Most of the studies that evaluated different categories of duration observed an elevated BC risk for long-term NSW (≥15/≥20 years) [13,14,15,16,17,18,19], in contrast to the moderate [16,17,19,20] or absent excess risk observed for shorter exposures (<15 years) [12,14,15,18,21,22].

Another important issue that might help to understand the inconsistent results is the possible role of menopausal status in this presumptive association. Pre- and postmenopausal breast cancer have different temporal trends, can differ in their molecular profile, and even have opposite relationships with some of the known risk factors for this tumor, such as obesity (i.e., obese women have a lower risk of premenopausal cancer and higher risk of postmenopausal tumors than women of normal weight) [23,24,25].

Another possible reason for these discrepancies might be related to a critical point: the lag time between the last NSW exposure and the end of follow-up for the occurrence of BC in some cohort studies [26,27]. Some authors have hypothesized that NSW may act essentially as a promoter in BC; if that were the case, when NSW exposure ceases, the associated BC risk increase should wane [13,28], while the increased risk associated with NSW should be mostly visible in recently exposed women. This pattern of decreasing risk since the cessation of exposure has also been proposed for the suggested association between NSW and other tumors [29]. As far as we know, no meta-analysis has focused on this interesting issue, although a pooled analysis of case–control studies found a higher premenopausal BC risk in women with recent exposure than in those who had stopped night work more than two years before [30].

For this reason, we carried out an overall meta-analysis of observational studies on long-term NSW and BC, a dose–response meta-analysis, and a sub-meta-analysis on recent long-term NSW. Therefore, this meta-analysis allows exploring both factors, the duration of NSW, and recent exposure in BC risk assessment. Given the possible biological differences between pre- and postmenopausal BC associated with long-term NSW, we also carried out subgroup analyses by menopausal status.

## 2. Materials and Methods

### 2.1. Literature Search and Eligibility Criteria

This review was conducted according to the PRISMA guidelines [31]. The protocol is registered in PROSPERO [32] (CRD42018102515), where methodological details can be consulted. We updated the search for original articles in PubMed/MEDLINE, Embase, and the Web of Science, using the terms (“nightshift” OR “night shift” OR “shiftwork” OR “shift work” OR “nightwork”) combined with “Breast cancer” from inception until 31 December 2020.

Our final search strategy was complemented with a manual search based on the references cited in the papers selected for full-text review. Additionally, we also went through the reference lists of the published meta-analyses. We took into consideration results in English, Spanish, German, or French. For this review, we considered eligible all original case–control, nested case–control, and cohort studies that compared, as risk exposure, long-term NSW (defined as ≥15 years) to having never done NSW. These studies had to provide original data and to have female BC risk as the outcome.

### 2.2. Data Extraction

Two members of our research team (A.P., C.S.) performed the original abstract/title screening, the full-text review, and the data extraction. In case of doubts or disagreements, a third author’s opinion (B.P.G.) allowed reaching a consensus. In the presence of diverse options of risk estimators, we used preferentially those including NSW exposure between 12.00 p.m. and 3.00 a.m. In cohort studies with several reports, we used the latest update if there were new cases or if the analysis was improved in terms of methodology. With respect to exposure ascertainment, we prioritized self-reported individual data over job-exposure matrix estimations. According to our inclusion criteria, we selected studies providing data corresponding to a minimum of 15 years of NSW, without restricting the exposure to a minimum number of nights per month. Authors from two studies were contacted to request additional data about the definition of nonexposure (information received) and mean age at BC diagnosis (information finally not available).

### 2.3. Risk of Bias Assessment of Individual Studies (Newcastle–Ottawa Scale)

The risk of bias of each study was evaluated by two researchers (A.P., C.S.) using the Newcastle–Ottawa scale (NOS) [33], an eight-item scale developed to assess the quality of nonrandomized studies for meta-analyses with a maximum punctuation of nine stars (see Appendix A). It assigns four stars to items related to selection and representativeness of the study groups. Another three stars correspond to the evaluation of the ascertainment of the exposure or the outcome of interest for case–control or cohort studies, respectively, and some of these items require operative definitions. In this case, for cohort studies, we assigned one star to those with at least 15 years of follow-up (15 years of NSW exposure does not determine the follow-up time, since exposure may be assessed retrospectively) and another to those with a difference ≤15% in the proportion of subjects lost to follow-up between exposed and unexposed. For case–control studies, we gave one star to those with differences ≤15% in nonresponse rates between cases and controls. Finally, the remaining two stars evaluate the comparability of the groups in terms of control for potential confounders; for this report, one star was given to studies that controlled for age and, additionally, for at least four of a predefined set of covariables (namely, family history of BC, age at first birth or number of children, body mass index in postmenopausal women, hormone replacement therapy, and menopausal status), and a second star was rendered to those studies that included at least five of these additional covariables: alcohol intake, oral contraception, age at menarche, age at menopause, breastfeeding, physical activity, history of benign breast disease, and education/socioeconomic status.

### 2.4. Statistical Methods

#### 2.4.1. Meta-Analysis on Long-Term NSW and BC Risk

Global meta-analysis of long-term NSW (for ≥15 years) and BC

We compared BC risk for women exposed to NSW for ≥15 years to that of unexposed women. We estimated summary risk ratios (RRs) and 95% confidence intervals (CIs) as the measure of effect. Individual effect estimates were extracted to construct a database with the logarithm of the risk estimator reported in each study (RR, hazard ratio (HR), or odds ratio (OR)) and its standard error (SE). If SEs were not provided in the publication, we calculated them from the CI, using the formula: SE = (lnCIupper limit-lnCIlower limit)/(2 × 1.96). When the report provided different categories of exposure but not an overall estimator, we calculated pooled RRs and variances using the formulas: lnRR pooled = (lnRR1/Var1 + lnRR2/Var2 + lnRRn/Varn)/(1/Var1 + 1/Var2 + 1/Varn) and Varpooled = 1/(1/Var1 + 1/Var2 + 1/Varn). The proportion of variance due to heterogeneity was estimated by the I^2^ statistic [34]. Heterogeneity between studies was assumed, and a random-effects model was applied using the method of DerSimonian and Laird, with the estimate of heterogeneity taken from the inverse-variance fixed-effects model.

To assess publication bias, funnel plot asymmetry was visually inspected and statistically tested with Begg’s and Egger’s tests (NFL, CS). We also carried out two sensitivity analyses: (a) excluding those studies with uncertain exposure definition (see Appendix A) and (b) restricting the analysis to those studies with NOS ≥ 7 (see Appendix A).

Program Codes Specific STATA^©^ version 15 codes (metan, metafunnel, metabias) from StataCorp. 2017. Stata Statistical Software: Release 15. College Station, TX: StataCorp LLC were used for the meta-analysis and RevMan 5.4 test for differences in the predefined subgroup analyses [35].

Dose–response meta-analysis

For studies reporting disaggregated results by duration of NSW, we further conducted a random-effects dose–response meta-analysis of log-relative risks (either adjusted log hazard ratios for cohort studies or adjusted log odds ratios for case–control studies) on mean durations of NSW for each exposure category [36]. The dose–response meta-analysis was fitted through generalized least squares, which allowed for within-study correlations among log-relative risks that used the same unexposed group as reference [37]. Linear trends were allowed to vary randomly across studies, whose variance was estimated by the method of moments [38]. We evaluated departures from the linear trend by including a quadratic term for mean duration, as well as distinct linear trends in cohort, nested case–control, and case–control studies by including interaction terms between mean duration and study design indicators. The heterogeneity of log-relative risks with respect to the pooled linear trend was contrasted with the Cochran chi-squared test and quantified with the I^2^ statistic, which described the proportion of total variation in log-relative risks due to heterogeneity [34]. The dose–response meta-analysis was performed using the glst command in Stata, version 14 from StataCorp. 2015. Stata Statistical Software: Release 14. College Station, TX: StataCorp LP, and graphics were produced in R, version 3 (R Foundation for Statistical Computing).

Subgroup meta-analysis by menopausal status

We stratified the overall analysis by menopausal status, which was preferentially based on self-reported information. If this information was not available, given cut-off points between age 50 and 55 were accepted to classify women either as pre- or as postmenopausal. Again, a sensitivity analysis was performed, restricting the meta-analyses to those studies with NOS ≥ 7.

#### 2.4.2. Meta-Analysis on Recent Long-Term NSW and BC

As the information on the timing of long-term NSW exposure was not readily available in the selected studies, we used an indirect approach based on the analysis of BC before retirement. We classified as having recent NSW those participants in the subgroup of studies in which, either the mean age of the participants at study entry plus the maximum follow-up time (cohort studies) or the mean age at BC diagnosis (case–control studies) was below the country-specific retirement age in that year; this age was obtained from the official government websites of each country or of the Organization for Economic Co-operation and Development [39]. The meta-analysis was performed following the same methodology that has been previously described.

## 3. Results

### 3.1. Search Results

The initial search yielded 1524 records found in PubMed, Embase, and Web of Science. After the selection process, 16 articles with data from 18 research studies were finally included in the global meta-analysis (Figure 1).

It is noteworthy to indicate that the early publications of the Nurses Health cohort studies (NHS I and II) [20,40,41] were excluded from the main meta-analysis, due to the availability of updated data with improved analyses from these projects, and that five studies included in previous meta-analyses did not meet our inclusion criteria, as they did not investigate long-term exposures (≥15 years of NSW) [28,42,43,44,45]. Our manual search did not identify any additional articles.

### 3.2. Study Characteristics

Table 1 summarizes the main characteristics of the studies included in our global meta-analysis, and the characteristics of other studies that did not meet our inclusion criteria but that were included in previous reviews; this way, the reader may have a rapid overview of the differences among the published meta-analyses in this area. In regard to those studies fulfilling our criteria, eight research projects, published in six articles, had a cohort design [12,13,14,22,46,47], five were nested case–control studies [17,18,21,48,49], and five were case–control studies [15,16,19,50,51]. The numbers of participants and BC cases in the table correspond to the risk estimators used for the present global meta-analysis. For risk estimators, all cohort studies reported HRs, except for one that reported RRs [20]; four of the five nested case–control studies reported ORs, and the remaining one reported HR [48], while all case–control studies reported ORs. Regarding NSW exposure, it should be noted that most of the studies only reported exposure status at baseline and did not update this information; therefore, the real number of women who might have finally been exposed to long-term NSW could be higher than reported, had we taken into account the additional years of NSW during follow-up.

Table 2 provides further details on exposure characteristics for the studies selected in this review, as well as indicates which are included in each of the meta-analyses presented in this report. The table summarizes the definition of exposure and nonexposure in each study. It should be noted that the operational approach used in several cases may raise doubts about the accuracy of the classification [14,21,22,47]. Thus, in the two Nurses’ Health Studies [14], the definition for nonexposure was “never rotating NSW”, with unclear classification for those women with permanent NSW. In another study among Norwegian nurses [49], any woman working at infirmaries was classified as exposed to NSW, while women working in managerial, teaching, physiotherapy, or at out-patients’ departments formed the nonexposed group. Additionally, in some reports only those with current NSW and long-term stability at the same job (“job tenure”) [22] or with NSW in the last 10 years [47], were considered as exposed, classifying as nonexposed those women with former NSW exposure.

Concerning the timing of exposure ascertainment, in cohort studies, it was mostly performed only at baseline, with just a few studies collecting data of additional years of NSW during follow-up [14,47]. In nested case–control studies and case–control studies, the ascertainment was retrospective, taking place after follow-up in the former and at recruitment in the latter. Self-reporting was the most common source of information to define NSW [12,13,14,15,16,17,18,19,47,49,50,51], while three studies [21,22,48] applied job-exposure matrices (JEM) for this purpose. Interestingly, the Shanghai Women’s Health cohort study compared both approaches, concluding that the use of JEM overestimated the prevalence of NSW [46].

As regards the duration of exposure, in 10 studies, the risk estimates provided by the authors corresponded to NSW exposures of at least 20 years [12,13,14,15,17,22,47,48,50] and in one study to exposures greater than 17 years [46].

### 3.3. Results of the Risk of Bias Assessment in the Selected Studies

The detailed risk of bias assessment for each selected study, according to the Newcastle–Ottawa scale (NOS), is available in the Appendix A. In summary, NOS scores ranged from four [22] to eight stars [14,16,18]. Among cohort studies, the highest scores were obtained by NHS I and II [14], penalized only by their focus on a single occupation. The other cohort studies studied the general population, except for one, which included twins [13]. All but three cohort studies [22,28,46] achieved our highest quality criteria for control of confounders. Only NHS I and II reported the proportion of participants lost to follow-up for exposed and nonexposed groups and had a follow-up length over our threshold [14], but two other cohort studies presented an overall number of losses to follow-up below 5% [46,47]. Three of the nested case–control studies received high scores in the NOS (≥7) [17,18,49], and all case–control studies reached at least six points. The highest score among them corresponded to a Canadian study [16], only penalized for not blinding the interviewer, a common problem that often has a difficult logistic solution in this type of study; in fact, only one case–control study used interviewers blinded to the disease status [50]. The other common quality problem present in case–control-studies [15,19,50] laid on differences in the response rate between cases and controls.

### 3.4. Results of the Meta-Analyses

#### 3.4.1. Long-Term NSW and BC Risk

Global meta-analysis on long-term NSW and BC risk

The meta-analysis of long-term NSW and BC risk (Figure 2) showed a global significant increased risk of BC, although there was moderate heterogeneity of effects between studies (pooled effect estimate = 1.13, 95%CI = 1.01–1.27, 18 studies, I^2^ = 56.8%, *p* = 0.002). When we took into account study designs, the meta-analysis yielded no association between NSW and BC in cohort studies (pooled effect estimate = 1.04, 95%CI = 0.91–1.18, I^2^ = 36.3%, *p* = 0.139); for nested case–control studies, it showed an almost significant risk increase, although with high heterogeneity (pooled effect estimate = 1.34, 95%CI = 0.96–1.88, I^2^ = 82.2%, *p* < 0.001); and, finally, in case–control studies, the pooled risk indicated an excess risk of BC among exposed women (pooled effect estimate = 1.22, 95%CI = 1.01–1.48, I^2^ = 0.0%, *p* = 0.613). We performed sensitivity analyses: (a) excluding the studies with uncertain exposure definition, which showed similar results (pooled effect estimate = 1.17, 95%CI = 0.99–1.39, I^2^ = 61.8%, *p* = 0.002) (Appendix A); and (b) including only studies that scored ≥7 stars at the NOS, which yielded the same overall risk estimate for long-term NSW and BC (pooled effect estimate = 1.13, 95%CI = 1.00–1.28, I^2^ = 48.2%, *p* = 0.031) (see Appendix A).

The funnel plot to evaluate a possible relation between the RR of BC and the study precision showed an imbalance (Figure 3), which was statistically significant (Egger’s test *p* = 0.03, Begg’s test *p* = 0.08). The asymmetry was not present in a funnel plot limited to the high-quality studies (NOS ≥ 7) (see Appendix A) (Egger’s test *p* = 0.33, Begg’s test *p* = 0.24).

Dose–response meta-analysis

The dose–response meta-analysis was based on 16 studies. Risk trends across categories of exposure duration varied substantially from one study to another, although most of them tended to show an increase in risk with rising duration of NSW (Figure 4). The pooled risk for BC increased by 4.7% for every 10 years of NSW duration (pooled effect estimate = 1.05, 95%CI = 0.94–1.09), with moderate heterogeneity across studies (I^2^ = 33.4%, *p* = 0.008), and no significant differences among cohort, nested case–control, and case–control studies (*p* for interaction between exposure duration and study design = 0.45). Adding a quadratic term to the model for exposure duration did not significantly improve model fit (*p* = 0.389).

Subgroup analysis by menopausal status

Menopausal status was generally self-reported, although in some studies, this information was not available. Overall, long-term NSW was associated with an almost significant increased risk for pre-/perimenopausal BC (pooled effect estimate = 1.27, 95%CI = 0.96–1.68, six studies, I^2^ = 32.0%, *p* = 0.196) (Figure 5), mostly due to the results of cohort studies, while no association was found for postmenopausal BC (pooled effect estimate = 1.05, 95%CI = 0.90–1.24, seven studies, I^2^ = 52.4%, *p* = 0.050) (Figure 6), although the test for subgroup differences was not significant (*p* = 0.27). However, in our sensitivity analysis, restricted to high-quality studies, the difference between both subgroups was more marked (pooled effect estimate _premenopause_ = 1.57; 95%: 1.11–2.21; pooled effect estimate _postmenopausee_ = 1.08; 95%: 0.94–1.24; test for subgroup differences: *p* = 0.05) (see Appendix A).

#### 3.4.2. Recent Long-Term NSW and BC Risk

This last meta-analysis included data from 15 studies. Among them, data from the NHS1 after 10 years of follow-up could be included, as the biological age of menopause normally is below the retirement age of 65 years [20]. The updated HRs for women from the NHS2 after 10 years of follow-up [14] and the HRs for women Shanghai textile workers under 50 years [48] could also be included. Moreover, we included a Danish nested case–control study [17], which provided mean age at the beginning of the study, and the sum of the years of follow-up resulted in ages below the retirement age in Denmark at that time. The meta-analysis of recent long-term NSW and BC showed a significant pooled risk of BC of around 23% (95%CI = 1.06–1.42), with moderate heterogeneity between studies (I^2^ = 48.4%, *p* = 0.018) (Figure 7) (test of interaction between NSW and BC before and after retirement: *p* = 0.001). The results were similar in all study design groups (20% for cohort studies, 95%CI = 0.88–1.65, 29% for nested case–control studies, 95%CI = 0.96–1.73, and 22% for case–control studies, 95%CI = 1.01–1.48).

### 3.5. Comparison with Former Meta-Analyses

Table 3 shows the main summary effect estimates of previous meta-analyses, dose–response meta-analyses, and sub-meta-analyses for the association between long-term NSW and BC risk, globally and in pre- and postmenopausal women, to allow the reader to have a better perspective of the results that have been published in this matter.

## 4. Discussion

### 4.1. The Association between Long-Term NSW and BC Risk

The global meta-analysis showed that the risk of BC in women exposed to ≥15 years of NSW is 13% higher than that of those who had not worked nightshifts, although there was moderate heterogeneity among the studies. In the dose–response meta-analysis, the BC risk increased by 4.7% for every 10 years of exposure to NSW. The analysis by menopausal status suggested that this association was limited to premenopausal BC, while no association was observed for BC risk after menopause; this difference was quite more marked when we restricted the analysis to high-quality studies. We also found an excess of risk among those women exposed to recent long-term NSW, but due to the scarcity of studies, we did not evaluate differences by menopausal status in this subgroup. It should be noted that the use of aggregated data on age and retirement as a proxy for recent exposure, due to the lack of individual information, renders it very difficult to separate the possible effects of the time lag (between exposure cessation and outcome) from those of menopausal status and age.

### 4.2. Comparison with Former Meta-Analyses and Relevant Studies

Our global findings are similar to the results of a meta-analysis carried out by Jia et al. [10], who reported a 15% increased BC risk for long-term NSW. Additionally, He et al. [54] found a 6% increased BC risk for every 10 years of exposure in their dose–response meta-analysis, similar to our result. Considering only the results from cohort studies, our pooled estimate was comparable with the null results of previous meta-analyses [12,54], while another one detected an 8.8% risk increase after 20 years of NSW exposure [11]. For premenopausal BC and long-term NSW, we found a slightly lower increased risk than the one reported by Cordina-Duverger et al. [30] for an exposure of at least twenty years after pooling the results of case–control studies, though in that study, no elevated risk was observed for exposure durations between 10 and 20 years. As this review includes, to our knowledge, the first intent of carrying out a sub-meta-analysis of BC risk before retirement and long-term NSW, we could not compare the results related to recent NSW exposure.

### 4.3. Interpretation of the Results and Possible Mechanisms of the Association

Our analyses of long-term NSW suggest that this exposure was associated with BC before retirement or menopause. It is conceivable that, during regular NSW exposure, changes in sex hormone secretion [65] or other biological mechanisms might drive the growth of malignant tumor cells with a promoter effect; if that were the case, once exposure ceases, the BC risk might drop to normal levels. Similar effects had been proposed for the action of hormone replacement therapy [66] but were later partly questioned by a meta-analysis [67]. Alternatively, some cancer-relevant epigenetic, genetic, or microenvironmental changes caused or favored by long-term NSW might contribute to transforming breast epithelial cells into malignant cells before menopause, supported by the higher cell proliferation rates at this stage [68]. It is also plausible that the NSW-related risks persist during more years, but cumulative DNA damage with age, caused by a wide spectrum of reasons [69], could mask the deleterious effects of NSW in older women. In regard to long-term NSW, having been working in night shifts for at least 15 years makes it more likely to be exposed in windows of increased susceptibility, that is, in life periods during which harmful factors can more easily lead to cancer. One of these susceptibility periods is the time before a first birth [70]. The early life etiological model of BC [70] suggests that childbirth and breastfeeding cause differentiation of mammary stem cells so that they are consequently less prone to proliferate and cause BC. Thus, NSW exposure after the first childbirth is expected to be less harmful than before this life event. However, the few risk estimates for NSW exposure before the first childbirth found in the literature are contradictory [19,45,50]. Menopausal transition might also be a period of higher susceptibility [71], but unfortunately, the impact of NSW during this susceptibility window has not been studied yet.

### 4.4. Strengths and Limitations

The exhaustive literature search, including studies published up to December 2020, allowed us to provide updated estimates of risk associated with long-term NSW, as well as to perform an updated dose–response meta-analysis. In contrast to a previously published report [30], we included cohort, nested case–control, and case–control studies. For this review, we assessed the risk of bias of included studies and carried out a sensitivity analysis based on their quality scores. Additionally, we explored the consistency of our results after excluding those studies with uncertain exposure definitions or ascertainment. However, probably, the major strength of this meta-analysis resides in its deeper analysis of the risk for long-term NSW, thanks to the sub-meta-analyses carried out by menopausal status, a factor involved in clinical and epidemiological differences in BC, and by recent exposures, which allow exploring the hypothesis under debate of a reduction of risk after the cessation of NSW exposure. Noteworthy, the subgroup analysis by menopausal status was mainly based on within studies comparisons. Our approach on recent long-term NSW, even with its mentioned limitations, represents an original analysis trying to overcome the scarcity of information about time since NSW cessation. Although we are not able in this review to decipher the reasons for the association found between long-term NSW and breast cancer during working age, we think it is an interesting new finding that deserves further investigation.

Notwithstanding, this review also has limitations. The asymmetry of the funnel plot may suggest that our estimates can be affected by publication bias or by differences in the quality of smaller studies that may spuriously inflate their effects [72]. However, our sensitivity analysis, restricted to the studies of higher methodological quality (NOS ≥ 7), yielded the same overall risk as the global meta-analysis, while asymmetry, according to the funnel plot and the Egg test, was no longer significant. Another limitation comes from the heterogeneity observed in the meta-analysis. As previously mentioned, we explored several putative factors that might contribute to explaining it (e.g., study design and study quality, menopause, exposure definition); however, other design characteristics, such as NSW pattern (e.g., permanent versus rotating shift work or frequency of shifts) or NSW assessment (i.e., self-report versus JEM), may also contribute to it. It has been described that self-report can be used as a reliable method for NSW exposure [73], while, in contrast, JEM might result in misclassification bias towards the null [46]. As the three included studies [21,22,48] that used JEMs received high weights in the meta-analysis (in total 23%), it is possible that we might be underestimating the risk due to NSW. The distribution of BC molecular subtypes might also differ among studies, being an additional uncontrolled potential source of heterogeneity, although it is not yet known whether NSW may have a different role in the pathogenesis of BC per subtype. Cordina-Duverger recently described higher risks of NSW linked to tumors that were both HER+ and ER+ [30]. Finally, we could only carry out an indirect approach to the risk associated with recent long-term NSW. This approach assumes that NSW is current or recent if the woman—really, the mean age of the participants—has not reached retirement age. However, we do not have data to test this assumption, which surely includes a certain misclassification. Additionally, it is difficult to disentangle the effect of menopause, as both are age related.

## 5. Conclusions

Our meta-analysis supports the hypothesis that long-term NSW is a risk factor for BC. Even though the possible effect of the publication bias is small, we prudently classify the evidence as low. Among the factors that may modify this risk, which are possible sources of the observed heterogeneity, are menopause and time since last NSW exposure. Both factors point towards higher risks in younger women. In contrast, no association was found between long-term NSW and BC in postmenopausal women. Future investigators on NSW and this tumor should also take into consideration the timing of exposure and the menopausal status at BC diagnosis.

## Figures and Tables

**Figure 1 cancers-13-05952-f001:**
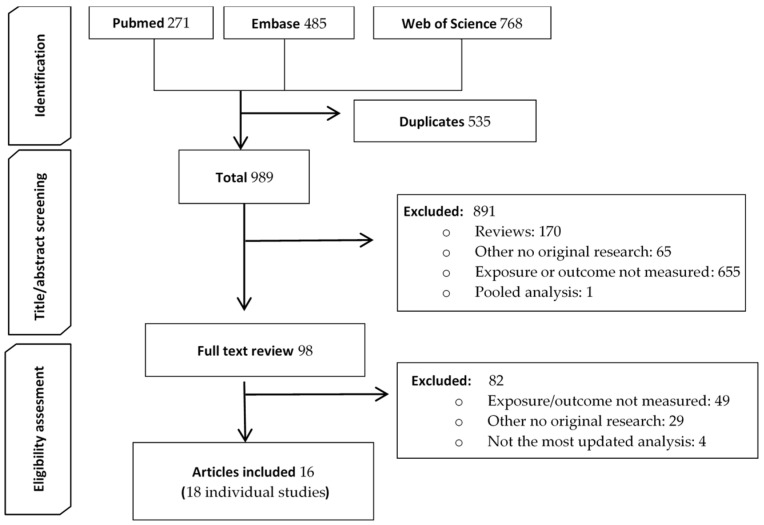
Study flow diagram.

**Figure 2 cancers-13-05952-f002:**
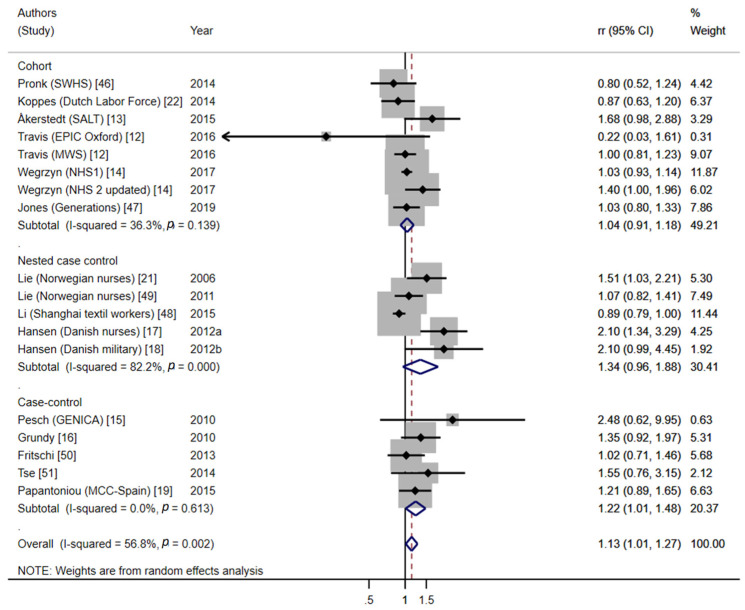
Forest plot of the meta-analysis on long-term nightshift work (≥15 years of NSW) and BC risk.

**Figure 3 cancers-13-05952-f003:**
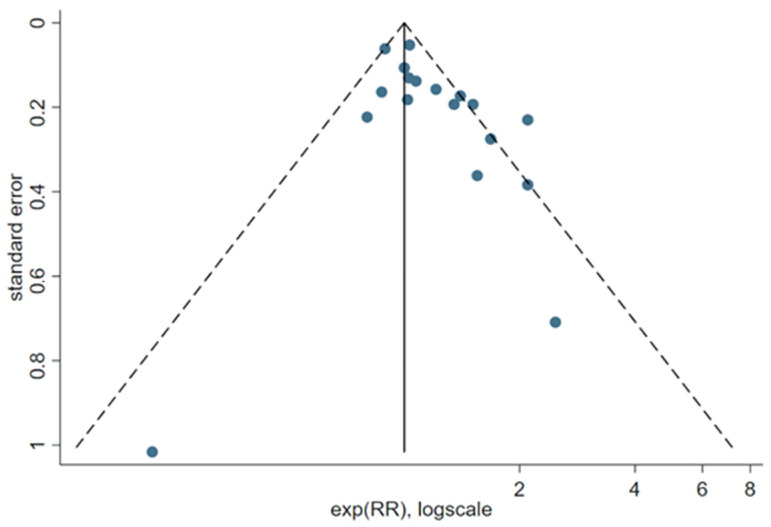
Funnel plot of reports included in the meta-analysis of long-term NSW and BC risk. Dashed lines represent pseudo 95% confidence limits.

**Figure 4 cancers-13-05952-f004:**
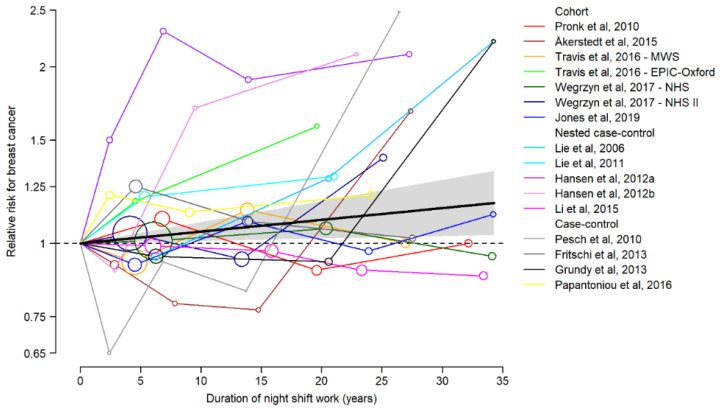
Dose–response meta-analysis with BC risk trends across categories of NSW duration. (For this analysis, the exposure categories of 10–20 years and ≥20 years of the EPIC-Oxford study were combined). Black line and shadowed area represent overall pooled RR and 95%CI.

**Figure 5 cancers-13-05952-f005:**
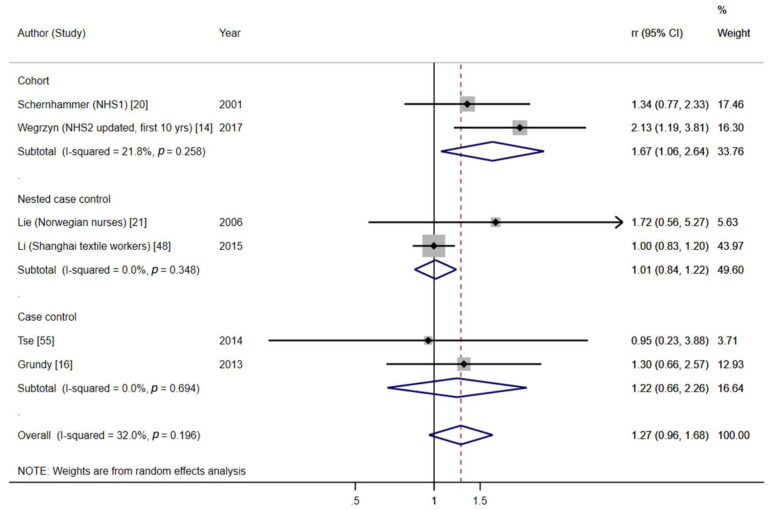
Forest plot of the meta-analysis on long-term NSW (≥15 years) and pre-/perimenopausal BC.

**Figure 6 cancers-13-05952-f006:**
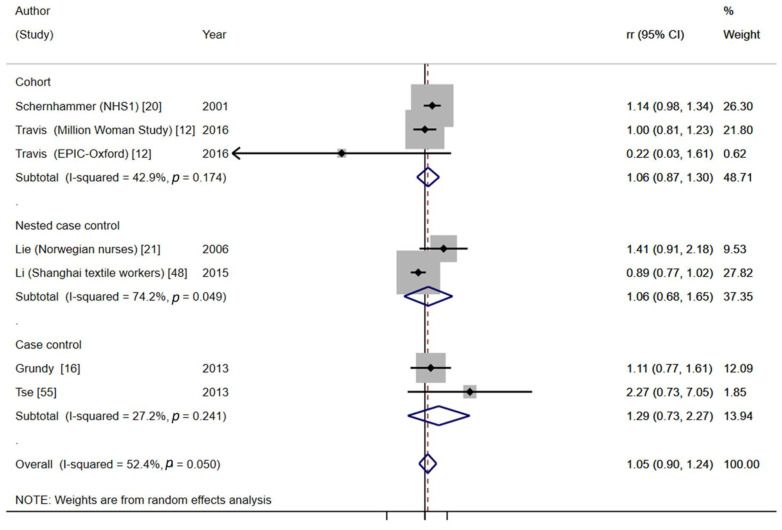
Forest plot of the meta-analysis on long-term NSW (≥15 years) and postmenopausal BC.

**Figure 7 cancers-13-05952-f007:**
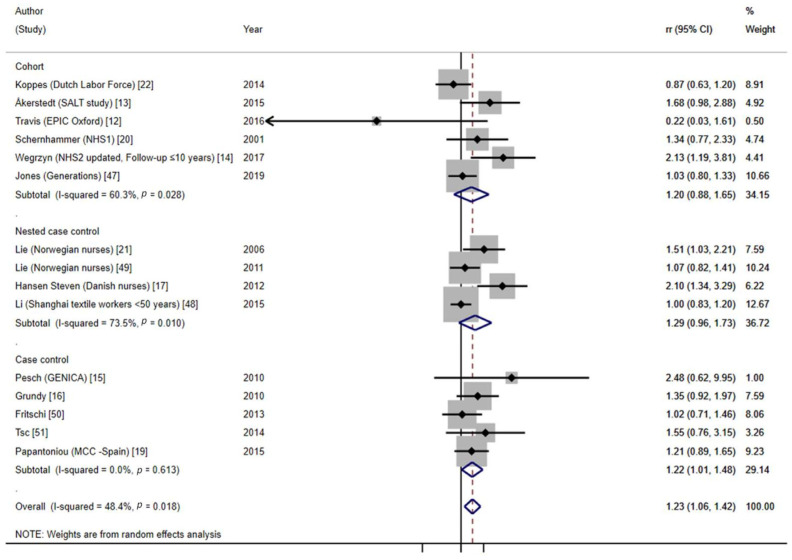
Forest plot of the meta-analysis on recent long-term NSW (≥15 years) and BC risk.

**Table 1 cancers-13-05952-t001:** Main characteristics of studies included in this or in previous meta-analyses and pooled analyses on long-term NSW exposure and BC.

(a) Cohort Studies																							
Individual Studies: Main Characteristics												Published Meta-Analyses or Pooled Analyses on Long-Term NSW											
												Wang (2013) [52]	Jia (2013) [10]	Ijaz (2013) [53]	He (2015) [54]	Lin (2015) [11]		Travis (2016) [12]		Cordina Duverger (2018) [30]		Present Meta-Analysis	Present Dose–Response Meta-Analysis
Author, Study, Year of Publication	Country	Base Population	Ascert. of Breast Cancer	≥15 yrs NSW		No NSW		Total Cases^5^	Follow-Up (yrs)	Lost to Follow-Up	Q	D-R per 5 yrs	≥15 yrs	D-R per 5 yrs	D-R per 10 yrs	10–20 yrs	>20 yrs	>20 yrs	>30 yrs	10–20 yrs	>20 yrs	≥15 yrs	D-R per 10 yrs
				N ^1^	Ca ^2^	N ^3^	Ca ^4^																
Schernhammer NHS I (2001) [20]	USA	Occup (nurses)	SR/HOSP						max 10	<10%	7	X	X	X	X	X	X	X	X	--	--	--	--
Wegrzyn NHS I (2017) [14]	USA	Occup (nurses)	SR/HOSP	5804	427	31,746	2382	2809	max 24	<10%	8	--	--	--	--	--	--	--	--	--	--	X	X
Schernhammer NHS I and II (2014) [41]	USA	Occup (nurses)	SR/HOSP						max 22	<10%	8	--	--	--		--	X	--	--	--	--	--	--
Schernhammer NHSII (2006) [40]	USA	Occup (nurses)	SR/HOSP						max 12	<10%	7	X	X	X	X	X	X	X	--	--	--	--	--
Wegrzyn NHS II (2017) [14]	USA	Occup (nurses)	SR/HOSP	162	35	43,529	950	985	max 24	<10%	8	--	--	--	--	--	--	--	--	--	--	X	X
Pronk (2010) SWHS based on JEM [46] ^6^	CHN	General pop	REG						av 9.0	3%	6	X	X	--	X	X	--	--	--	--	--	--	--
Ponk (2010) SWHS based on SR [46]	CHN	Working pop	REG	5720	19	51,238	276	295	av 4.4	3%	7	--	--	X	--	--	--	X	X	--	--	X	X
Knutsson WOLF (2013) [28]	SE	General pop	REG						av 12.4	n/a	6	--	--	--	--	X	--	--	--	--	--	--	--
Koppes Dutch Labor Force (2014) [22]	NL	Working pop	REG	n/a	n/a	255,900	2312	n/a	av 6.9	n/a	4	--	--	--	--	X	X	X	--	--	--	X	--
Åkerstedt Salt Study (2015) [13]	SE	Twins	REG	305	18	9674	354	372	av 8.7	n/a	6	--	--	--	--	--	--	X	--	--	--	X	X
Travis (2016) EPIC—Oxford [12]	UK	General pop	REG	461	1	19,289	153	154	av 3.1	n/a	7	--	--	--	--	--	--	X	--	--	--	X	X
Travis Million Women Study (2016) [12]	UK	General pop	REG	9647	89	450,232	4136	4225	av 2.6	n/a	7	--	--	--	--	--	--	X	X	--	--	X	X
Jones Generations Study (2019) [47]	UK	General pop	REG	n/a	60	84,888	1845	1905	median 9.5	4%	7	--	--	--	--	--	--	--	--	--	--	X	X
**(b)** **Nested Case-Control and Case-Control Studies**																							
**Individual Studies: Main Characteristics**												**Published Meta-Analyses or Pooled Analyses on Long-Term NSW**											
												**Wang** **(2013)** [52]	**Jia** **(2013)** [10]	**Ijaz** **(2013)** [53]	**He** **(2015)** [54]	**Lin** **(2015)** [11]		**Travis** **(2016)** [12]		**Cordina** **Duverger** **(2018)** [30]		**Present** **Meta-Anal.**	**Present Dose–Response Meta-Anal.**
	**Country**	**Base** **Population**	**Control** **Selection**	**Case** **Ascert**	**Ratio**	**LT-NSW Exposed (n)**		**Participation (%)**			**Q**	**D-R** **per** **5 yrs**	**≥** **15** **yrs**	**D-R** **per** **5 yrs**	**D-R** **per** **10 yrs**	**10–20** **yrs**	**>20** **yrs**	**>20** **yrs**	**>30** **yrs**	**10–20** **yrs**	**>20** **yrs**	**≥** **15** **yrs**	**D-R** **per** **10 yrs**
						**Controls**		**Cases**	**Controls**	**Cases**													
NESTED CASE–CONTROL STUDIES																							
Tynes (1996) [42]	NOR	Occup (radio/tele)	n/a	REG	4–7:1				n/a	n/a	5	--	--	X	--	--	--	--	--	--	--	--	--
Lie (2006) [21]	NOR	Occup (nurses)	IDS	REG	4:1	417		125	n/a	n/a	5	X	--	X	--	--	--	--	--	--	--	X	X
Lie (2011) [49]	NOR	Occup (nurses)	IDS	REG	1.5:1	231		179	65	74	7	X	X	X	X	--	--	--	--	--	--	X	X
Hansen (2012a) [17]	DK	Occup (nurses)	IDS	REG	4:1	124		39	91	92	7	X	--	X	X	--	--	--	--	--	--	X	X
Hansen (2012b) [18]	DK	Occup (military)	IDS	REG	4:1	29		12	61	67	8	X	X	X	X	--	--	--	--	--	--	X	X
Li (2015) [48]	CHN	Occup (textile)	RND	REG	2.8:1	n/a		576	n/a	n/a	5	--	--	--	--	--	--	X	X	--	--	X	X
CASE–CONTROL STUDIES																							
Davis (2001) [43]	USA	General pop	RND	REG	1:1				75	78	5	X	--	X	X	--	--	--	--	--	--	--	--
O’ Leary (2006) LIBCSP/EBCLIS [44]	USA	Diverse sources	RND	HOSP	1:1				83	87	5	X	--	X	X	--	--	--	--	--	--	--	--
Pesch GENICA (2010) [15]	DE	General pop	RND	HOSP	1:1	5		12	67	88	6	X	X	X	X	--	--	--	--	X	X	X	X
Menegaux (2013) [45]	FR	General pop	RND	HOSP	1:1				76	79	6	--	--	X	X	--	--	--	--	X	X	--	--
Fritschi (2013) [50]	AUS	General pop	RND	REG	1.5:1	53		84	41	58	7	--	--	--	X	--	--	--	--	X	X	X	X
Grundy (2013) [16]	CA	BC screening	RND	Mixed ^7^	1:1	53		65	V: 54, K: 49	V: 57 K:59	8	--	--	--	X	--	--	--	--	X	X	X	X
Tsc (2014 [51,55]	CHN	Hospitalized	CONSEC	HOSP	1:1	n/a		n/a	93 (55)	91 (55)	6	--	--	--	--	--	--	--	--	--	--	X	--
Papantoniou (2015) [19]	SP	General pop	RND	HOSP	1:1	97		91	52	72	7	--	--	--	--	--	--	--	--	X	X	X	X

Ascert (ascertainment), yrs (years), Q (quality index of the Newcastle–Ottawa scale; score range: 0–9), D-R (dose–response meta-analysis), Occup (occupational), SR (self-reported), SR/HOSP (self-reported/hospital), max (maximum), SWHS (Shanghai Women’s Health Study), pop (population), REG (registry), av (average), n/a (not available), Ratio (ratio controls–cases), LT-NSW (long-term nightshift work), radio/tele (radio/telegraph operators), IDS (incidence density sampling), RND (randomly), V (controls and cases from Vancouver), K (controls and cases from Kensington), CONSEC (consecutive) ^1^ Number of women exposed to ≥15 yrs of NSW. ^2^ BC cases among women exposed to ≥15 yrs of NSW. ^3^ Number of women not exposed to NSW. ^4^ Number of BC cases among women not exposed to NSW. ^5^ Total number of cases included in the present meta-analysis. ^6^ This study used a job-exposure matrix and self-reporting to ascertain NSW exposure. ^7^ Registry + BC screening.

**Table 2 cancers-13-05952-t002:** Exposure characteristics and age of participants in relation to retirement age in each study country, in the included publications.

(a) Cohort Studies																
Author, Year of Publication, Study	Exposure Definition	Min. Freq. of NSW/Month	Nonexposure Definition	Exposure Ascertainment		NSW Duration (yrs)	Age at Start of Follow-Up		Follow-Up Period	Mean Age at Start of Follow-Up + Full Follow-Up Period (yrs)	Retirement Age (yrs)	Estimated yrs since Retirement at the End of Follow-Up	Inclusion in our (sub)-Meta-Analyses			
				Direction and Moment	Source		Mean	Range					All BC	Premen BC	Postmen BC	“Recent Long-Term NSW”
Schernhammer (2001), NHS I [20]	Yrs of rotating night shifts with at least 3 nights/month	3/m	Never rotating NSW ^1^	Retro. at start of follow-up	SR	≥15	57.1	42–67	1988–1998	67.1	65 [56]	2.1	--	X	X	X ^2^
Wegrzyn (2017), NHS I [14]	See Schernhammer (2001)	3/m	Never rotating NSW ^1^	Retro. at start of follow-up	SR	≥15	57.1	42–67	1988–2012	81.1	65 [56]	16.1	X	--	--	--
Wegrzyn (2017), NHS II update [14]	See Schernhammer (2001)	3/m	Never rotating NSW ^1^	Retro. at start of follow-up and 5 updates	SR	≥20	39	25–42	1989–2013	63.0	66 [56]	Still working, but no NSW	X	X	--	X ^3^
Pronk (2010) SWHS [46]	Starting work after 10 p.m.	3/m	Never night shifts	Retro. and two yrsafter start of follow-up	SR	>17	52.5	40–70	2000–2007	61.5	50–55 [57]	6.5–11.5	X	--	--	--
Koppes (2014), Dutch Labor Force [22]	Occ/regular current work at night, between 0 and 6 a.m. combined with yrs of job tenure	None	No current NSW≥20 yrs same job	Before start of follow-up	JEM	≥20	38 *	15–64	1996–2009	51.9	65 [58]	Still working	X	--	--	X
Åkerstedt (2015), Salt twin study [13]	Yrs working at nights at least now and then	None	Not worked nights	Retro. at start of follow-up	SR	21–45	51.6	41–60	1998–2010	63.6	65 [59]	Still working	X	--	--	X
Travis (2016), EPIC-Oxford [12]	Worked regularly at night, on NSW or on call at night	1/m	Never night shifts	Retro. at start of follow-up	SR	≥20	57.8	n/a	2010–2013	60.9	62 [60]	Still working	X	--	X	X
Travis (2016), MWS [12]	Ever regularly worked at night or on night shifts at any time between 00 and 06 h	3/m	Never night shifts	Retro. at start of follow-up	SR	≥20	68.6	n/a	2011–2013	71.2	60 [60]	11.2	X	--	X	--
Jones (2019), Generations Study [47]	Any job that regularly involved work in the late evening or night (between 10 p.m. and 7 a.m.)	None	Not being an NSWer within the last ten yrs	At start of follow-up and 1 update 6 yrs after recruitment	SR	≥20	45 **	35–55	2003–2018	60.5	60–68 [60]	Still working	X	--	--	X
**(b) Nested Case-Control and Case-Control Studies**																
**NESTED CASE CONTROL STUDIES**																
**Author,** **Year of Publication**	**Exposure Definition**	**Min. Freq of** **NSW/month**	**Non-Exposure** **Definition**	**Source of Expo. Ascert**		**NSW** **Duration** **(yrs)**	**Age at Start of Follow-Up**		**Mean Age at dx of BC (yrs) [Range]**	**Mean Age at Start of Follow-Up + Full Follow-Up Period (yrs)**	**Retirement** **Age (yrs)**	**Estimated** **yrs since** **Retirement at BC dx**	**Inclusion in our (sub) Meta-Analyses**			
							**Mean**	**Range**					**All BC**	**Premen** **BC**	**Postmen** **BC**	**“Recent Long-Term NSW”**
Lie (2006) [21]	Work at infirmaries (based on Nurse Registry and census, only considering time after grad)	None	Managerial, teaching, physiotherapy, outpatient department worksite, other than infirmaries	JEM		15–29	39.7 *	27–85	54 * (27–85)	61.7	67 [61]	Still working	X	X	X	X
Lie (2011) [49]	Rotating and permanent night work	3/m	Never night work	SR		≥15	n/a	20–70	54.5 (35–74)	n/a	67 [61]	Still working	X	--	--	X
Hansen (2012a) [17]	Working ≥1 yr during hours between 7 p.m. and 9 a.m. not including overtime	None	Never (<1 year) “after midnight shifts”	SR		≥20	54 *	31–69	n/a	56	65 [62]	Still working ***	X	--	--	X
Hansen (2012b) [18]	Working ≥1 yr during hours between 5 p.m. and 9 a.m., not including overtime	None	No yrs (<1 yr) of NSW	SR		≥15	n/a	22–75	n/a	n/a		n/a	X	--	--	--
Li (2015) [48]	Jobs involving rotating NSW (22.00–06.00 h) according to factory processes	None	No rotating NSW ^4^	JEM		20–27	48.9 *	30–66	53.4	59.9	50 [57]	3.4	X	X	X	X ^5^
**CASE CONTROL STUDIES**																
**Author** **Year of publication, study**	**Exposure Definition**	**Min.** **freq. of NSW/Month**	**Nonexposure Definition**	**Source of expo. Ascert**		**NSW Duration** **(yrs)**			**Mean Age at dx of BC (yrs) [Range]**	**Year of** **BC- dx**	**Retirement** **Age (yrs)**	**Estimated yrs since Retirement at BC Diagnosis**	**Inclusion in our (sub) Meta-Analyses**			
													**All BC**	**Premen** **BC**	**Postmen** **BC**	**“Recent Long-Term NSW”**
Pesch (2010) [15]	Ever having worked in NSW for ≥1 year and working the full time period between 0.00 and 5.00	None	Employed but never in shift work (day shifts only)	SR		≥20			54 (42–62)	2000–2004	65 [63]	Still working	X	--	--	X
Fritschi (2013) [50]	Worked any number of hours between 0.00 and 5 a.m. (graveyard shift)	None	Never graveyard shift	SR		≥20			57 * (18–80)	2009–2011	64.5 [63]	Still working	X	--	--	X
Grundy (2013) [16]	Jobs that started/ended between 11 p.m. and 7 a.m.	None	No yrs in jobs with start or end between 11 p.m. and 7 a.m.	SR		≥15			57 (<80)	2005–2010	65 [39]	Still working	X	X	X	X
Tsc (2014) [51,55]	Nightshift at least once per month for ≥1 year	1/m	Permanent day work ^6^	SR		≥15			55 (40–69)	2011–2013	65 [64]	Still working	X	X	X	X
Papantoniou (2015) MCC-Spain [19]	Partly/entirely working between 0 and 6 a.m.	3/m	Never night work	SR		≥15			56 (23–85)	2008–2013	65 [39]	Still working	X	--	--	X

Min (minimum), freq (frequency), yrs (years), Retirement age (legal retirement age in the study country at the moment of the study), Premen (premenopausal), Postmen (postmenopausal), m (month), retro (retrospective), SR (self-reported), NSWer (nightshift worker), Occ (occasionally), JEM (job-exposure matrix), n/a (not available), dx (diagnosis), yrs (years), expo ascert. (exposure ascertainment), dx (diagnosis), grad (graduation). ^1^ May include permanent NSW. ^2^ Only the RR for premenopausal was included. ^3^ The first ten years of follow-up were included, as no NSW was assumed since 2009 by study authors. ^4^ No jobs in the factory involved permanent NSW. ^5^ The BC risk for women < 50 years was included. ^6^ Kindly reported by e-mail from the author, * Calculated based on given categories of age or birth year; ** Median age; *** Mean age at BC diagnosis was missing in this study, but the mean age at start of follow-up + years of full follow-up was under the retirement age.

**Table 3 cancers-13-05952-t003:** Summary results of the meta-analyses, dose-response meta-analyses and sub-meta-analyses for the association between long-term NSW and BC risk.

Meta-Analysis	Wang (2013) [52]	Jia (2013) [10]	Ijaz (2013) [53]	He (2015) [54]	Lin (2015) [11]	Travis (2016) [12]	Cordina-Duverger (2018) [30]	Present Study
**Inclusion** **Criteria** **(Study Types)**	Cohort, nested case–control, and case–control	Cohort, nested case–control and case–control	Cohort, nested case–control, and case–control	Cohort, nested case–control and case–control	Cohort	Cohort and nested case–control	Case–control	Cohort, nested case–control, and case–control
**Duration of exposure**								
**10–20 years**					1.07 (1.01–1.14)		0.98 (0.78–1.22)	
**≥15 years**		1.15 (1.03–1.29)						1.13 (1.01–1.27)
**≥20 years**					1.09 (1.01–1.17)	1.01 (0.93–1.10)	1.10 (0.87–1.39)	
**≥30 years**						1.00 (0.87–1.14)		
**Dose–response meta-analysis**								
**per 5 years**	1.03 (1.01–1.05)		1.05 (1.01–1.10)					
**per 10 years**				1.06 (0.98–1.15)				1.05 (0.94–1.09)
**Sub-meta-analyses**								
** *Premenopausal women* **								
**10–20 years**							1.05 (0.74–1.47)	
**≥15 years**								1.27 (0.96–1.68)
**≥20 years**							1.34 (0.85–2.13)	
** *Postmenopausal women* **								
**10–20 years**							0.92 (0.68–1.23)	
**≥15 years**								1.05 (0.90–1.24)
**≥20 years**							1.04 (0.80–1.36)	

## Data Availability

Publicly available datasets were analyzed in this study. The analyzed data can be found in the reported publications (see references of Table 2).

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
