# Peer review of "Long-Term Nightshift Work and Breast Cancer Risk: An Updated Systematic Review and Meta-Analysis with Special Attention to Menopausal Status and to Recent Nightshift Work"

_cancers, 2021, doi:10.3390/cancers13235952_

Round 1

Reviewer 1 Report

The current systemic review " Long-term Nightshift Work and Breast Cancer Risk: An updated systematic review and meta-analysis with special attention to menopausal status and to recent nightshift work" assessed the correlation between long term night shifts and breast cancer risks. They compared the cohort studies and showed the correlation which is same as shown by other studies. Though the study is increment, but provides a better understanding as statistical analysis is performed in a better way. Conclusion were made in realistic way and also talks about the strength and limitation of the study. Overall the study is conducted in scientifically stablished manner and presented well.

Author Response

Thank you very much for your review, as well as for your favourable appreciation on the quality and on the statistical analysis of our work. 

Reviewer 2 Report

Review for article cancers-1430620

Long-term Nightshift Work and Breast Cancer Risk: An updated systematic review and meta-analysis with special attention to menopausal status and to recent nightshift work

            This meta-analysis is performed very correctly and in accordance with the PRISMA guidelines. The authors put a lot of work into analyzing all the data. The study is well conducted and the methodology is solid. The topic is very interesting considering that many women around the world work night shifts.

I feel to addressed the following minor points:

  1. On page 2 (Introduction, Line 98): it is “before [27]” and it should be “before [27].”
  2. In the supplementary, above the table is "Table S.1" and it should be "Table S1".
  3. On page 4 (Materials and Methods, Line 150): it is “not available -, body” and it should be “not available - body”.
  4. The format should change from Figure 5.
  5. The font size should be changed in line 332-333 for Figure 5.
  6. For the data in Figure 5, Figure 7, Figure S1 and Figure S2, I would include literature references (easier to read).
  7. Please see reference [42 - Pronk].Figure S1 should say 2010, not 2014.
  8. On page 13 (Line 346): it is “(Figure 7)” and it should be “(Figure 7)” (bold).

Author Response

Thanks a lot for these kind comments, and for the recognition of our efforts in this review.  We are happy to see that you consider our methodology as solid and that you have enjoyed our work.  

Thank you for this careful review. We have corrected all the points/ format changes and misspellings that you indicate.

Reviewer 3 Report

The authors investigated the association between long term nightshift work (NSW) and breast cancer in a meta-analysis of 18 case-control and cohort studies. Additional secondary analyses included investigations of duration of NSW, pre/post menopausal status effect modification, and recent NSW.

  1. There is no justification in the introduction as to why menopause would be an expected effect modifier.
  2. Formal statistical interaction tests should be performed for the menopause and recent NSW effect. Please see Sun X, et al. How to use a subgroup analysis: users’ guide to the medical literature. JAMA 2014;311(4):405-411. In short, please interact pre/post menopause with long term NSW on breast cancer risk (and also do this for recent/not NSW). I am not convinced that the results differ statistically for premenopausal women compared to postmenopausal women, but this should be formally tested.
  3. The novelty is not clear. Based on Table 3, nothing new is provided. We already know from the meta-analysis by Jia 2013 that long term NSW increases breast cancer risk. The meta-analysis by He 2015 provides a risk estimate per 10 years of NSW. The meta-analysis by Cordina-Duverger 2018 provides estimates by menopausal status. Finally, the results regarding recent NSW are quite flawed as noted by the authors in 4.1.
  4. The results are overstated in the abstract and discussion. The risk estimate per 10 years of NSW and estimate for pre-menopausal women are not statistically significant.
  5. The substantial heterogeneity and funnel plot do not lend confidence to these results. It is a comment that I do not think can be overcome.

Author Response

REVIEWER 3

The authors investigated the association between long term nightshift work (NSW) and breast cancer in a meta-analysis of 18 case-control and cohort studies. Additional secondary analyses included investigations of duration of NSW, pre/post-menopausal status effect modification, and recent NSW.

Thanks for your review. Your comments have helped us to improve the manuscript. Also, we hope to have satisfied your concerns with our additional analyses and explanations,

1. There is no justification in the introduction as to why menopause would be an expected effect modifier.

Thank you for signalling us this point. The evaluation of the effect of risk factors by menopausal status is very usual in breast cancer research, as some have features such as obesity, have opposite effects depending on it, and pre- and postmenopausal tumours may have different characteristics.  Therefore, we intended to do this analysis since we started the review. We have added a paragraph explaining in more detail the rationale for the possible modifying effect of menopausal status, as follows:

Another important issue that might help to understand the inconsistent results is the possible role of menopausal status in this presumptive association. Pre and post-menopausal breast cancer have different temporal trends, can differ in their molecular profile, and have even opposite relationships with some of the known risk factors for this tumour, such as obesity (i.e. obese women have lower risk  of  premenopausal cancer and higher risk of postmenopausal tumours than women of normal weight  [23–25].

2. Formal statistical interaction tests should be performed for the menopause and recent NSW effect. Please see Sun X, et al. How to use a subgroup analysis: users’ guide to the medical literature. JAMA 2014;311(4):405-411. In short, please interact pre/post menopause with long term NSW on breast cancer risk (and also do this for recent/not NSW). I am not convinced that the results differ statistically for premenopausal women compared to postmenopausal women, but this should be formally tested.

The protocol of our review pre-specified selected subgroup analyses (menopause, recent work) based on previous literature -including biological support- and on our hypotheses. Following your recommendation, we have included formal statistical test of interaction in the analysis for menopause for the main meta-analysis as well as in the sensitivity analysis, restricted to high quality data. While in the first case, the interaction term is not statistically significant, in the second, which shows higher differences between both groups, the interaction test p was 0.05. All these points, which are detailed in the manuscript, are well aligned with the criteria to evaluate subgroup analyses stated in the reference that you mention.

3. The novelty is not clear. Based on Table 3, nothing new is provided. We already know from the meta-analysis by Jia 2013 that long term NSW increases breast cancer risk. The meta-analysis by He 2015 provides a risk estimate per 10 years of NSW. The meta-analysis by Cordina-Duverger 2018 provides estimates by menopausal status. Finally, the results regarding recent NSW are quite flawed as noted by the authors in 4.1.

As we explain in the manuscript, the exhaustive literature search, including studies published up to December 2020, has allowed us to provide updated estimates of risk associated to long-term NSW, in the light of the new evidence that has been published since some of the papers that the reviewer mentions. as well as to perform an updated dose- response meta-regression analysis (we include 4 new studies). In contrast to a previously published report [30], our approach was comprehensive, and included cohort, nested case control and case control studies. For this review, we assessed the risk of bias of included studies, and carried out a sensitivity analysis based on their quality scores. Also, we explored the consistency of our results after excluding those studies with uncertain exposure definitions or ascertainment. However, probably, the major strength of this meta-analysis resides in its deeper analysis of the risk for long-term NSW, thanks to the sub-meta-analyses carried out by menopausal status, a factor involved in clinical and epidemiological differences in BC, and for recent exposures, which allow to explore the hypothesis under debate of a reduction of risk after the cessation of NSW exposure. Our approach on recent long-term NSW, even with its mentioned limitations, represents an original analysis trying to overcome the scarcity of information about time since NSW cessation. Although we are not able in this review to decipher the rea-sons for the association found between long-term NSW and breast cancer during working age, we think it is an interesting new finding that deserves further investigation.

4. The results are overstated in the abstract and discussion. The risk estimate per 10 years of NSW and estimate for pre-menopausal women are not statistically significant.

Following your advice, we have reformulated the conclusions in the abstract and discussion more cautiously. However, statistical significance should not be the only criteria used, and we have evaluated these results in light of the previous evidence and taking into account our sensitivity analyses, which reinforce the hypothesis that NSW may have a different effect by menopausal status.

Concerning the risk estimate of the dose response meta-analysis, chance can surely also explain the reported effect. However, we were testing whether BC risks increases with a longer duration of exposure, a hypothesis based on former studies.  Our analysis update those reported by HE (2015), as we include four new studies that they could not take into account, and the results in both are very similar, suggesting that there might be a duration dependent effect of NSW.

5. The substantial heterogeneity and funnel plot do not lend confidence to these results. It is a comment that I do not think can be overcome.

We understand your concern about these very relevant issues. In regard to the funnel plot results, we have included in the manuscript an additional funnel plot corresponding to the meta-analysis of those studies of higher methodological quality (NOS ≥ 7), which was already present in the previous version of this paper as sensitivity analysis. This last meta-analysis yielded the same overall risk than the global meta-analysis, while asymmetry, according to the funnel plot and the Egg test, was no longer significant. These data suggest certain consistency in our results. We have included the following text among the study limitations:

The asymmetry of the funnel plot may suggest that our estimates can be affected by publication bias or by differences in quality of smaller studies that may spuriously inflate their effects [72]. However, our sensitivity analysis, restricted to the studies of higher methodological quality (NOS ≥ 7), yielded the same overall risk than the global meta-analysis while asymmetry, according to the funnel plot and the Egg test, was no longer significant.

In regard to the other point you raise, it is true that the global meta-analysis presents a relevant heterogeneity, and, following your remark, we have remarked this in the first paragraph of the discussion, and in the conclussion. However, the aim of the sub-meta-analysis we present in this manuscript is, precisely, to deepen into the understanding of this heterogeneity. This is the reason why we present the results by type of study, and explore the main possible sources of variability, such as study quality, menopause or exposure definition.  However, there are additional factors that we could not evaluate, and we cannot combine all of them in a single analysis due to the low number of studies. The heterogeneity, in fact, really summarizes that there is variability among the selected studies, and is not derived from quality problems in our search or approach.  We have explained this in detail in the text.

Another limitation comes from the heterogeneity observed in the meta-analysis. As previously mentioned, we have explored several putative factors that might contribute to explain it (e.g., study design and study quality, menopause, exposure definition); however, other design characteristics, such as NSW pattern (e.g., permanent versus rotating shift work or frequency of shifts) or NSW assessment (i.e., self-report versus JEM) may be also contributing to it. It has been described that self-report can be used as a reliable method for NSW exposure [73], while, in contrast, JEM might result in misclassification bias towards the null [46]. As the three included stud-ies [21,22,48] that used JEMs received high weights in the meta-analysis (in total 23%), it is possible that we might be underestimating the risk due to NSW.  The distribution of BC molecular subtypes might also differ among studies, being an additional uncon-trolled potential source of heterogeneity, although it is not yet known whether NSW may have a different role in the pathogenesis of BC per subtype. Cordina-Duverger re-cently described higher risks of NSW linked to tumours that were both HER+ and ER+ [30].

Round 2

Reviewer 3 Report

My comments have been adequately addressed.